# UNIVDC: A ZERO-SHOT UNIFIED DIFFUSION FRAMEWORK FOR CONSISTENT VIDEO DEPTH COMPLETION

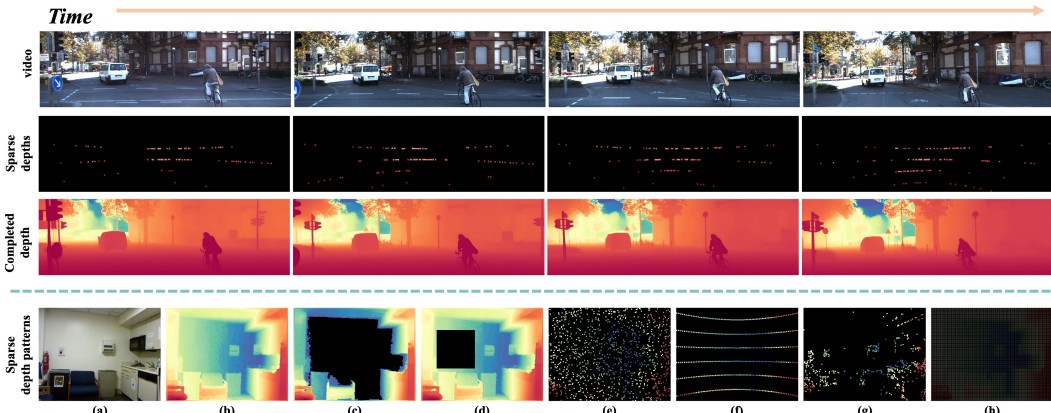

Figure 1: **UniVDC**—the first zero-shot unified video diffusion framework that jointly completes sparse depth and inpaints structural degradations. From diverse sparse/damaged inputs (top, rows 1–2), UniVDC yields metrically consistent and temporally coherent depth (row 3). Bottom: varied sparse depth patterns all supported by one model via a four-stage training protocol and bidirectional overlapping sliding-window Inference (BOSW) inference.

## ABSTRACT

Recovering metrically consistent and temporally stable depth from dynamic videos remains challenging, particularly when sparse, noisy measurements co-exist with structural voids, occlusion reveals, motion drift, and sensor dropouts. Under these conditions, single-frame methods lack temporal correction while existing video depth estimation approaches underutilize explicit sparse geometry, leading to scale drift and flicker. To address this, we introduce UniVDC, the first unified zero-shot spatiotemporal diffusion framework for long-range video depth completion. Our approach centers on multi-source geometric and semantic priors. We combine two geometric inputs: fine-grained relative depth with structural and edge cues from a depth estimator, and coarse metric depth obtained by inverse-distance–weighted interpolation of sparse measurements. Unlike methods that feed RGB frames directly, we extract global semantic features and inject them hierarchically into the diffusion network, yielding compact geometric inputs and scene context robust to frame-level appearance noise. A four-stage training protocol stabilizes prior fusion and calibrates the long-horizon scale. In inference, we introduce bidirectional overlapping sliding-window (BOSW) to reduce scale drift and boundary error accumulation over long sequences and alleviate occlusion in one-directional inference. Experiments show that UniVDC achieves state-of-the-art performance on multiple zero-shot video depth completion benchmarks in terms of completion accuracy, structural consistency, and temporal coherence.

## 1 INTRODUCTION

Depth is fundamental to 3D perception tasks across autonomous navigation (Carranza-García et al., 2022; Häne et al., 2017; Tao et al., 2022), robotics (Bagnell et al., 2010; Kim & Chen, 2015), video generation (Zhang et al.; 2021) and AR/VR (Rasla & Beyeler, 2022; Slater et al., 1997; Holynski & Kopf, 2018). In realistic video streams, sparse, irregular, and noisy LiDAR/SfM (Schonberger & Frahm, 2016)/RGBD (Silberman et al., 2012a)/structured-light (Lange & Seitz, 2001; Herrera et al., 2012) samples appear alongside occlusion reveals, block voids, motion or exposure drift, and sensor dropouts, producing large spatiotemporal holes beyond classical inpainting. Completing depth that is geometrically continuous, semantically coherent, and temporally consistent under globally sparse yet locally degraded observations remains a significant challenge. However, existing single-frame depth completion methods (Viola et al., 2024; Liu et al., 2024; Zuo et al., 2024; Wang et al., 2025; Lin et al., 2025) suffer from parallax ignorance and lack temporal self-correction mechanisms. Similarly, current video depth estimation approaches (Hu et al., 2025; Yang et al., 2024a; Chen et al., 2025; Shao et al., 2025; Wang et al., 2023) rely primarily on relative cues while underexploiting explicit sparse measurements. They smooth or cache features yet propagate errors unidirectionally, leading to amplified long-range scale drift and flicker. Therefore, a unified methodology capable of fully leveraging sparse geometry and temporal context while maintaining robustness to heterogeneous degradations is imperative.

To bridge this gap, we propose UniVDC: the first zero-shot unified diffusion framework for long-range video depth completion and inpainting that explicitly elevates temporal context to a co-primary structural prior alongside sparse geometry, relative depth, and semantics. Our framework employs a unified spatiotemporal diffusion backbone conditioned on four complementary priors: (1) coarse sparse-guided depth maps that supply local geometric anchors; (2) robust relative depth cues regulating global ordinal relationships and scale consistency; (3) temporally aggregated semantic features synchronizing texture-geometry alignment; and (4) explicitly injected temporal context, enhanced through a bidirectional sliding-window mechanism. To ensure robust gradient flow while disentangling multi-source priors, we implement a four-stage training protocol: initial single-frame warm-up progresses to short-range joint spatiotemporal propagation, followed by decoupled temporal-spatial stabilization stages. This hierarchical refinement progressively locks in structural coherence and metric stability. During inference, our bidirectional overlapping sliding-window (BOSW) strategy executes concurrent forward-backward diffusion with adaptive overlap fusion. This design mitigates unidirectional scale drift, boundary inconsistencies, and localized flicker artifacts. It also enable seamless generalization across sparse completion, structural repair, and hybrid tasks, all without task-specific fine-tuning.

We distill our technical advances into three main contributions:

- The First Unified Video Diffusion with Four-Stage Training for Zero-Shot Depth Completion: We propose a video-oriented unified diffusion framework with a four-stage training strategy that jointly tackles sparse depth completion and structural damage inpainting, yielding generalized zero-shot performance across diverse sparsity patterns and scene domains.
- Bidirectional Overlapping Sliding-Window (BOSW) Inference: We devise a bidirectional, overlap-aware sliding window inference strategy that suppresses long-range scale drift and mitigates local flicker, strengthening temporal coherence over extended sequences.
- State-of-the-Art Accuracy and Consistency: Across diverse video depth completion and inpainting benchmarks, our method attains state-of-the-art performance in both completion fidelity and temporal/structural consistency.

## 2 RELATED WORKS

**Single-frame Depth Completion.** Depth completion fuses sparse or degraded depth with RGB to produce dense, scale-stable maps. Contemporary approaches (Zuo et al., 2024; Wang et al., 2025; Lin et al., 2025) integrate multi-scale priors or generative architectures. Diffusion or prior-enhanced methods (Viola et al., 2024; Liu et al., 2024) improve zero-shot robustness yet remain challenged by ultra-low sampling density, severe noise corruption, and long-sequence temporal inconsistency. Consequently, efficient high-resolution inference with guaranteed temporal stability remains an open challenge.

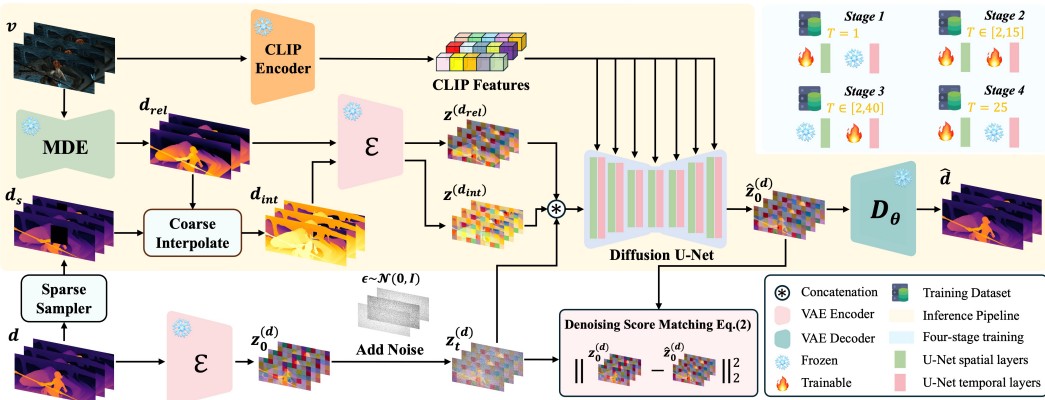

Figure 2: **Overall Pipeline of UniVDC.** Given video frames combined with arbitrary sparse or degraded depth ($d_s$) and monocular relative depth cues ($d_{rel}$), we VAE-encode the inputs—including the coarse depth completion ($d_{int}$)—to condition a spatiotemporal diffusion U-Net for recovering fine-grained, scale-consistent depth. After encoding, both the relative depth ($z^{(d_{rel})}$) and the coarse completion ($z^{(d_{int})}$) are used for conditioning. A four-stage training strategy increases sequence length and activates temporal and spatial layers progressively; BOSW inference (Figure 3) later fuses forward/backward windows for long-range stability.

**Monocular Depth Estimation.** Monocular depth estimation aims to infer relative geometry from RGB inputs; while large-scale pretraining and geometric regularization enhance generalization, single-frame models (Yang et al., 2024b;c; Bochkovskii et al., 2024; Birkl et al., 2023; Hu et al., 2024; Ke et al., 2024; Fu et al., 2024) lack persistent scale anchoring and exhibit limited occlusion coherence. Recent video-based (Blattmann et al., 2023a;b; Chen et al., 2024; Ho et al., 2022) or diffusion-based methods (Hu et al., 2025; Yang et al., 2024a; Chen et al., 2025; Shao et al., 2025; Wang et al., 2023) employ sliding windows, global aggregation, or feature reuse to mitigate temporal flicker. However, most rely on relative scaling and underutilize explicit sparse geometric constraints, fundamentally limiting metric consistency. Motivated by these limitations, we introduce UniVDC: a unified framework combining multi-scale sparse-semantic fusion with generative geometric priors and temporal consistency regularization for generalized depth completion across diverse sampling patterns and domains.

# 3 METHOD

As shown in Figure 2, given an in-the-wild video $\mathbf{v} \in \mathbb{R}^{T \times 3 \times H \times W}$, sparse or degraded depth $\mathbf{s} \in \mathbb{R}^{T \times 1 \times H \times W}$, and a validity mask $\mathbf{m} \in \mathbb{R}^{T \times 1 \times H \times W}$, we aim to recover a dense depth sequence $\mathbf{d} \in \mathbb{R}^{T \times 1 \times H \times W}$ with complete geometry, persistent cross-frame scale consistency, temporal stability, and degradation robustness. Here $T$ denotes the frame count, and $H$ and $W$ denote spatial dimensions. UniVDC integrates (1) multi-source geometric & semantic priors, (2) a four-stage training protocol, and (3) bidirectional overlapping sliding-window (BOSW) inference, jointly improving cross-domain generalization and long-range consistency.

## 3.1 FOUNDATIONS OF VIDEO DIFFUSION MODELING

We adapt Stable Video Diffusion (SVD) (Blattmann et al., 2023a) to depth completion by jointly encoding conditioning inputs (RGB + sparse-derived priors) and target depth into a temporal latent space (Rombach et al., 2022) to reinforce coherence and efficiency. The encoder $\mathcal{E}$ produces $\mathbf{z}^{(\mathbf{x})} = \mathcal{E}(\mathbf{x})$ (where $\mathbf{z}^{(\mathbf{x})}$ denotes the joint encoded representation of multi-source priors used for conditioning, $z(\mathbf{x}) = \mathcal{E}(d_{int} \oplus d_{rel}) + c_{sem}$), $\mathbf{z}_0^{(\mathbf{d})} = \mathcal{E}(\mathbf{d})$ (where $\mathbf{d}$ denotes the depth), and forward diffusion adds noise with noise level $\sigma_t$:

$$\mathbf{z}_t^{(\mathbf{d})} = \mathbf{z}_0^{(\mathbf{d})} + \sigma_t^2 \boldsymbol{\epsilon}, \boldsymbol{\epsilon} \sim \mathcal{N}(\mathbf{0}, \mathbf{I}). \quad (1)$$

Training then minimizes a denoising score matching (DSM) (Vincent, 2011) objective

$$\mathcal{L}_{depth} = \mathbb{E}_{\mathbf{z}^{(\mathbf{d})}, \mathbf{z}^{(\mathbf{x})}, \sigma_t} \left[ \lambda(\sigma_t) \| D_\theta(\mathbf{z}_t^{(\mathbf{d})}; \sigma_t, \mathbf{z}^{(\mathbf{x})}) - \mathbf{z}_0^{(\mathbf{d})} \|_2^2 \right] \quad (2)$$

Figure 3: **Bidirectional Overlapping Sliding-window Inference.** Forward and backward window streams with overlap O produce latent sequences. Overlapping frames employ linear ramp blending, while new windows initialize overlaps using prior clean latents augmented with Gaussian noise. After generating forward ($\hat{\mathbf{z}}^{(f)}$) and backward ($\hat{\mathbf{z}}^{(b)}$) latents, a global linear weighting $w_f(t), w_b(t) = 1 - w_f(t)$ fuses them into the final sequence, supplying early stabilization and posterior correction for long-range depth consistency.

with $\lambda(\sigma) = (1 + \sigma^2)\sigma^{-2}$ to balance gradients across noise scales. The denoiser $D_\theta$, a temporally aware U-Net (Ronneberger et al., 2015), employs EDM (Karras et al., 2022) preconditioning for numerical stability over wide noise ranges:

$$D_\theta(\mathbf{z}_t^{(\mathbf{d})}; \sigma_t, \mathbf{z}^{(\mathbf{x})}) = c_{\text{skip}}(\sigma_t)\mathbf{z}_t^{(\mathbf{d})} + c_{\text{out}}(\sigma_t)F_\theta(c_{\text{in}}(\sigma_t)\mathbf{z}_t^{(\mathbf{d})}; c_{\text{noise}}(\sigma_t), \mathbf{z}^{(\mathbf{x})}), \quad (3)$$

where $F_\theta$ is a learnable U-Net, and $c_{\text{skip}}, c_{\text{out}}, c_{\text{noise}}$ govern residual path modulation, output scaling, and noise encoding across noise levels.

## 3.2 MODEL DESIGN

We decouple conditioning into geometric and semantic streams: sparse metric depth anchors combined with monocular relative depth establish scale and topological constraints, while a global semantic vector provides scene-object context. A local inverse-distance interpolation corrected by a lightweight linear alignment to relative depth produces a coarse depth $\mathbf{d}_c$ that is continuous, scale-stabilized, and edge-aware. $(\mathbf{d}_c, \mathbf{d}_{\text{rel}})$ are fed into the diffusion, excluding raw RGB to prevent texture and illumination noise from amplifying cross-domain bias. Temporal semantic coherence is injected via separate embedding $\mathbf{c}_{\text{sem}}$ with minimal dimensionality overhead, prioritizing structural fidelity and robustness noise.

### 3.2.1 CONSTRUCTION OF GEOMETRIC AND SEMANTIC PRIORS

Prior construction proceeds through: relative depth inference, locally aligned coarse metric completion, robust percentile normalization, and finally injection of a global semantic embedding.

**Relative Depth Prior.** A pretrained monocular model (Depth Anything v2 (Yang et al., 2024c)) extracts $\mathbf{d}_{\text{rel}}$ encoding ordinal relationships and boundaries. Though metrically uncalibrated, it remains stable across appearance variations and complements sparse anchors: relative depth supplies global layout and sharp boundary continuities, while anchors restore absolute scale.

**Coarse Metric Depth Completion ("Coarse Interpolate" in Figure 2).** For each frame, observed depths $\mathbf{d}_{\text{prior}}^t$ are retained on valid pixels $\Omega_t$, then fill missing regions via inverse-distance weighted interpolation over $k$ nearest anchors:

$$\mathbf{d}_c^t(p) = \sum_{i=1}^k w_i \mathbf{d}_{\text{prior}}^t(p_i), \quad w_i = \frac{\sum_{j=1}^k d_j}{d_i}. \quad (4)$$

To mitigate over-smoothing, a lightweight local linear fit (scale/shift) aligns $\mathbf{d}_{\text{rel}}$'s neighborhood values to anchors. This fit is applied to missing pixels while valid pixels remain unchanged:

$$\mathbf{d}_c^t(p) = \begin{cases} \mathbf{d}_{\text{prior}}^t(p), & p \in \Omega_t, \\ s \cdot \mathbf{d}_{\text{rel}}^t(p) + t, & p \notin \Omega_t \end{cases}; s, t = \arg\min_{s,t} \sum_{i=1}^k w_i \| s \cdot \mathbf{d}_{\text{rel}}(p_i) + t - \mathbf{d}_{\text{prior}}(p_i)\|_2. \quad (5)$$

This hybrid approach (anchors + localized ordinal alignment) suppresses scale drift in large gaps and better preserves discontinuities than pure interpolation or global rescaling.

**Robust normalization.** Per-frame robust scaling is applied: clip both depth maps to their 1st–99th percentile range, linearly map to $[-1, 1]$, then concatenate $(\mathbf{d}_c, \mathbf{d}_{rel})$ as the unified geometric tensor. Clipping suppresses extreme outliers, thus stabilizing latent distribution statistics without RGB reliance.

$$\hat{d} = 2 \cdot \frac{\text{clip}(d, d^-, d^+) - d^-}{d^+ - d^-} - 1, \quad d^- = P_{1\%}(d), ; d^+ = P_{99\%}(d) \tag{6}$$

**Global Semantic Conditioning.** A frozen CLIP (Radford et al., 2021) encoder extracts a single aggregated semantic embedding $\mathbf{c}_{sem}$ from the video, which we inject it via conditioning layers instead of channel concatenation. This maintains lean geometry inputs while adding scene-object context uncontaminated by frame-level appearance noise.

### 3.2.2 Model Structure Adaptation and Scale Alignment

We freeze the VAE (Rombach et al., 2022) and semantic encoder, finetuning exclusively the diffusion U-Net (Ronneberger et al., 2015). Freezing preserves pretrained priors and confines optimization to geometry-temporal refinement.

**Model Input Channel.** The first-layer input is extended to incorporate $(\mathbf{d}_c, \mathbf{d}_{rel})$. Initial convolution weights are adjusted via channel ratio scaling to maintain activation variance, preserving pretrained spatial-temporal inductive bias while avoiding early feature saturation.

**Metric Depth Output.** The decoded unscaled depth $\mathbf{d}$ undergoes metric recovery through least-squares optimization (solving for global scale $s$ and shift $t$) over valid anchor pixels. The affine transformation $\mathbf{d}_{metric} = s\mathbf{d} + t$ is then applied globally, compensating residual scale drift from relative priors and stochastic sampling.

### 3.3 Training Protocol

UniVDC integrates: (1) multi-domain synthetic data, (2) unified degradation/sparsification, and (3) four-stage spatiotemporal training, targeting zero-shot depth completion with structural fidelity and temporal stability across domains.

**Datasets and Sampling.** Uniformly shuffled clips are sampled from five synthetic sources (Hypersim (Roberts et al., 2021), TartanAir (Wang et al., 2020), MatrixCity (Li et al., 2023), GTA5-540 (Huang et al., 2018), vKITTI2 (Cabon et al., 2020)) covering indoor realism, motion/climate diversity, urban scale hierarchy, traffic semantics, and controllable roadway layouts. Synthetic corpora provide precise metric depth/pose, programmable appearance diversity, and consistent scaling.

**Unified Degradation Synthesis.** As shown in Figure 1, exactly one degradation is applied per iteration to avoid supervision ambiguity from compounded artifacts and enforce modality-agnostic geometric priors. The figure depicts: (a) RGB reference; (b) full ground-truth depth; (c) range truncation (beyond-threshold removal); (d) rectangular occlusion; (e) sparse random points (minimal pixel retention); (f) simulated 8-/16-line LiDAR patterns; (g) keypoint-driven anchors; (h) 8× downsampling. Single-mode application ensures clearer gradients and reduces overfitting to composite degradations.

**Four-Stage Spatiotemporal Training Protocol.** Our four-stage strategy (Figure 2) isolates then progressively reintegrates spatial/temporal learning. Stage I (T=1) trains spatial modules exclusively, establishing robust local completion, boundary fidelity, and degradation-agnostic priors. Stage II (2-15 frames) engages temporal attention/normalization to learn correspondence, motion-aware aggregation, and cross-frame depth propagation for complementary hole filling. Stage III (2-40 frames) freezes spatial weights while refining temporal pathways to enforce long-range scale consistency, sustained occlusion recovery, and object persistence without degrading spatial embeddings. Stage IV (25 frames) re-enables lightly regularized spatial updates while freezing temporal parameters, sharpening high-frequency details and thin structures under stabilized temporal priors. This progression mitigates gradient interference, yielding globally coherent geometry with refined microstructure and enhanced cross-domain generalization.

## 3.4 BIDIRECTIONAL OVERLAPPING SLIDING-WINDOW

Existing long-video depth systems (Hu et al., 2025; Shao et al., 2025; Yang et al., 2024a) employ unidirectional sliding windows with causal conditioning. These approaches (1) propagate early scale drift/artifacts recursively and (2) prevents posterior frames from resolving earlier occlusions/holes—thus perpetuating structural biases. We propose Bidirectional Overlapping Sliding-Window Inference (BOSW), introducing a symmetric backward chain fused via lightweight operations to suppress directional error accumulation while preserving metric scale.

As shown in Figure 3, let $T$ denote the frame count, $W$ the window length, overlap $O$ ($0 < O < W$), stride $S = W - O$. Forward windows $\mathcal{W}_f$ advance head→tail; backward windows $\mathcal{W}_b$ traverse tail→head. Frame overlaps yield dual latent estimates per direction.

**Intra-direction local overlap fusion.** For overlapping forward windows $W_a = [s_a, e_a]$, $W_b = [s_b, e_b]$ with $s_b \leq t \leq e_a$, blend by $\alpha_t^{(f)} = \frac{e_a - t}{e_a - s_b}$, $\hat{\mathbf{z}}_0^{(f)}(t) = \alpha_t^{(f)} \mathbf{z}_0^{(a)}(t) + (1 - \alpha_t^{(f)})\mathbf{z}_0^{(b)}(t)$. Non-overlapped frames propagate unchanged. The backward chain mirrors this procedure, smoothing latent transitions and preventing seam artifacts.

**Final overlap fusion.** Overlap initialization injects the prior window's clean latent with variance-matched Gaussian noise, preserving scale consistency and edge anchors while maintaining stochastic diversity. After intra-direction blending yielding $\hat{\mathbf{z}}_0^{(f)}$ and $\hat{\mathbf{z}}_0^{(b)}$, we globally weight them by: $w_f(t) = 1 - \frac{t-1}{T-1}$, $w_b(t) = \frac{t-1}{T-1}$ ($w_f + w_b = 1$) forming $\hat{\mathbf{z}}_0^{(\text{final})}(t) = w_f(t)\hat{\mathbf{z}}_0^{(f)}(t) + w_b(t)\hat{\mathbf{z}}_0^{(b)}(t)$. Early frames optimize forward stability, mid-frames attenuate localized anomalies through symmetric processing, while late frames employ backward correction to enhance global consistency.

Posterior frames frequently reveal geometry obscured earlier (disocclusions, thin structures), and the backward chain provides this evidence, transforming unidirectional error propagation into bidirectional equilibrium. Linear overlap ramps ensure continuity, noise-anchored initialization balances structural anchoring with diversity, and global temporal weighting stabilizes metric scale. As Figure 4 and Figure 6 show, these components collectively suppress drift, enhance disocclusion recovery, and maintain boundary coherence in ultra-long depth completion.

## 4 EXPERIMENTS

### 4.1 EXPERIMENTAL SETUP

**Datasets.** We employ five benchmarks covering outdoor, indoor, dynamic, synthetic, and handheld settings: KITTI (Geiger et al., 2013) (sparse LiDAR, wide metric range), NYUv2 (Silberman et al., 2012b) (indoor static structure), BONN (Palazzolo et al., 2019) (fast human/object motion with occlusion cycles), Sintel (Butler et al., 2012) (large displacement, blur, appearance shift), and ScanNetV2 (handheld egomotion with repeated occlusion cycles). Collectively, they encompass scale variation, motion, occlusion, and domain shift.

**Degradation and Prior Configurations.** As shown in Figure 1, we follow PriorDA (Wang et al., 2025) and group settings as: (1) Completion: 8-line LiDAR, sparse SfM (ORB (Rublee et al., 2011)/SIFT (Lowe, 1999)) points, extremely sparse Bernoulli sampling. (2) Inpainting: range truncation ($\leq 2m$ indoor / $\leq 15m$ outdoor), random rectangles, resolution downsampling. (3) Mixed: composite of sparsity + masking + resolution loss. A unified masking interface tests adaptive fusion across heterogeneous inputs.

**Baselines.** To our knowledge, this is the first work formalizing unified video depth completion. We compare three lineages: (1) Single-frame depth estimators (Depth Anything V2 (Yang et al., 2024c), DepthPro (Bochkovskii et al., 2024)). (2) Video depth models (DepthCrafter (Hu et al., 2025), ChronoDepth (Shao et al., 2025), Depth Any VideoYang et al. (2024a), Video Depth Anything (Chen et al., 2025)). (3) Single-frame completion models (DepthLabLiu et al. (2024), Marigold-DC (Viola et al., 2024), Omni-DC (Zuo et al., 2024), PriorDA (Wang et al., 2025)). This contrasts prior utilization against temporal modeling. Relative-depth methods are metric-aligned via least-squares on valid prior pixels.

Table 1: **Zero-shot Depth Completion.** ScanNet-TAE reports TAE↓, all other results are reported in AbsRel↓. "S": points sampled with SIFT and ORB; "E": 100 random points; "L": 8 LiDAR lines.

| Model | KITTI | | | NYUv2 | | | Bonn | | | Sintel | | | ScanNet | | | ScanNet-TAE | | |
|---|---|---|---|---|---|---|---|---|---|---|---|---|---|---|---|---|---|---|
| | S | E | L | S | E | L | S | E | L | S | E | L | S | E | L | S | E | L |
| DAV2 | 13.02 | 12.72 | 12.84 | 30.39 | 29.21 | 29.02 | 9.13 | 8.32 | 8.12 | 38.45 | 45.96 | 46.16 | 13.15 | 12.65 | 12.61 | 2.217 | 2.254 | 2.253 |
| Depth pro | 12.15 | 9.89 | 10.37 | 26.97 | 20.19 | 20.33 | 6.73 | 6.39 | 6.18 | 112.18 | 130.51 | 111.77 | 9.31 | 8.7 | 8.51 | 3.207 | 3.328 | 3.321 |
| DepthCrafter | 11.48 | 10.92 | 10.56 | 27.97 | 29.73 | 28.83 | 7.08 | 6.41 | 6.33 | 50.29 | 41.85 | 38.11 | 13.56 | 13.23 | 13.18 | 1.776 | 1.91 | 1.829 |
| ChronoDepth | 21.42 | 17.77 | 18.56 | 36.92 | 34.46 | 34.42 | 8.52 | 7.99 | 7.81 | 55.44 | 70.59 | 69.26 | 16.45 | 16.28 | 15.97 | 1.532 | 1.525 | 1.528 |
| Depth Any Video | 8.61 | 8.1 | 7.82 | 16.61 | 17.14 | 17.25 | 8.16 | 7.18 | 6.02 | 37.82 | 42.5 | 46.26 | 10.98 | 10.68 | 10.45 | 1.763 | 1.898 | 1.885 |
| Video DA | 8.21 | 7.97 | 8.01 | 18.75 | 19.31 | 18.26 | 6.03 | 5.14 | 5.28 | **32.66** | **36.83** | 37.06 | 8.75 | 8.2 | 8.09 | 1.171 | 1.199 | 1.198 |
| DepthLab | 33.13 | 43.03 | 44.65 | 9.32 | 11.02 | 10.13 | 7.41 | 7.8 | 5.85 | 240.42 | 327.75 | 222.33 | 8.08 | 6.94 | 5.08 | 4.718 | 5.33 | 3.397 |
| Marigold-DC | 7.1 | 5.72 | 6.89 | 9.83 | 9.37 | 9.05 | 1.94 | 2.25 | 1.85 | 122.86 | 63.5 | 55.58 | 4.71 | 3.08 | 3.64 | 2.899 | 2.439 | 1.774 |
| Omni-DC | **4.87** | 4.31 | 5.4 | 8.81 | 8.1 | 7.93 | 1.83 | 2.6 | 1.97 | 40.1 | 64.02 | 40.27 | 4.64 | 2.83 | 3.21 | 2.575 | 2.621 | 1.545 |
| PriorDA | 5.27 | **3.63** | 4.46 | 8.83 | 8.89 | 8.83 | 2.38 | 2.27 | 2.27 | 63.78 | 60.07 | 70.74 | 4.48 | 2.76 | 3.33 | 1.991 | 1.783 | 1.446 |
| UniVDC(ours) | 5.11 | 4.21 | 5.11 | **8.52** | 8.78 | 8.45 | 3.06 | 2.14 | 2.92 | 36.77 | 41.39 | 35.72 | 4.25 | 2.98 | 3.17 | 1.032 | 0.875 | 0.928 |

Table 2: **Zero-shot Depth Inpainting.** ScanNet-TAE reports TAE↓, all other results are reported in AbsRel↓. "C": average result for random square masks; "R": masks for depth beyond 2m (indoors) and 15m (outdoors); "D": applying 8× downsampling to the GT depths.

| Model | KITTI | | | NYUv2 | | | Bonn | | | Sintel | | | ScanNet | | | ScanNet-TAE | | |
|---|---|---|---|---|---|---|---|---|---|---|---|---|---|---|---|---|---|---|
| | C | R | D | C | R | D | C | R | D | C | R | D | C | R | D | C | R | D |
| DAV2 | 12.76 | 14.75 | 13.18 | 29.26 | 37.29 | 28.91 | 8.42 | 20.67 | 8.45 | 44.14 | 45.98 | 45.15 | 12.66 | 15.58 | 12.64 | 2.26 | 1.898 | 2.244 |
| Depth pro | 9.84 | 13.69 | 9.98 | 19.98 | 28.91 | 20.16 | 6.38 | 19.84 | 6.39 | 136 | 52.56 | 130.21 | 8.69 | 11.16 | 8.73 | 3.332 | 2.932 | 3.318 |
| DepthCrafter | 11.04 | 11.12 | 10.83 | 30.2 | 38.98 | 29.38 | 6.49 | 19.61 | 6.43 | 38.98 | 39.88 | 39.15 | 13.22 | 16.46 | 13.21 | 1.898 | 1.528 | 1.828 |
| ChronoDepth | 17.55 | 17.05 | 17.5 | 34.43 | 38.37 | 34.35 | 8.14 | 18.29 | 8.15 | 70.71 | 37.91 | 70.79 | 16.27 | 19.04 | 16.35 | 1.527 | 1.315 | 1.525 |
| Depth Any Video | 8.12 | 10.74 | 8.87 | 16.95 | 26.44 | 16.72 | 7.03 | 19.94 | 7.47 | 43.06 | 47.88 | 44.86 | 10.6 | 12.98 | 10.37 | 1.887 | 1.782 | 1.876 |
| Video DA | 7.97 | 10.19 | 8.65 | 19.52 | 31.23 | 18.83 | 5.31 | 19.6 | 5.38 | 38.01 | 39.8 | **35.73** | 8.22 | 10.92 | 8.17 | 1.199 | 1.127 | 1.193 |
| DepthLab | 12.24 | 16.15 | 30 | 8.88 | 26.24 | 7.66 | 2.29 | 22.51 | 3.71 | 55.39 | 20.68 | 237.79 | **2.07** | 12.05 | 2.86 | 2.028 | 2.055 | 1.742 |
| Marigold-DC | 2.97 | 11.9 | 6.74 | 8.76 | 26.84 | 8.79 | 1.93 | 20.6 | 2.7 | 58.14 | 19 | 70.72 | 2.47 | 10.99 | 3.17 | 1.398 | 2.451 | 1.296 |
| Omni-DC | **1.66** | 11.23 | **5.49** | 9.06 | 26.27 | 7.71 | 2.36 | 20.53 | **1.5** | 68.26 | 18.66 | 37.05 | 2.59 | 11.67 | 2.4 | 2.519 | 2.545 | 0.931 |
| PriorDA | 2.42 | 13.74 | 6.03 | 9.2 | 26.57 | 8.19 | 2.57 | 22.35 | 2.87 | 48.81 | 19.66 | 48.29 | 2.41 | 12.08 | 2.72 | 1.212 | 2.302 | 0.908 |
| UniVDC(ours) | 3.88 | 10.56 | 5.99 | 8.75 | 10.94 | 7.97 | 1.86 | 19.47 | 2.09 | 36.71 | 18.27 | 49.88 | 2.33 | 9.88 | 2.61 | 0.932 | 1.115 | 0.856 |

**Evaluation Metrics.** For completion: Absolute Relative Error ($\frac{1}{N}\sum_{k=0}^{N-1}\frac{|\hat{x}_d - x_d|}{x_d}$) and $\delta_1$ ($\frac{1}{N}\sum_{k=0}^{N-1}\max(\frac{\hat{x}_d}{x_d}, \frac{x_d}{\hat{x}_d}) < 1.25$). Temporal Alignment Error (TAE) (Yang et al., 2024a) quantifies geometric consistency:

$$\text{TAE} = \frac{1}{2(T-2)}\sum_{k=0}^{T-1}\text{AbsRel}\left(f(\hat{x}_d^k, p^k), \hat{x}_d^{k+1}\right) + \text{AbsRel}\left(f(\hat{x}_d^{k+1}, p^{k+1}), \hat{x}_d^k\right) \quad (7)$$

where $p^{t+1}$ denotes reverse correspondence from $p^t$, enhancing sensitivity to scale drift and flicker.

**Training.** We employ DepthCrafter's (Hu et al., 2025) pretrained SVD architecture, trained on 8×A800 GPUs with mixed precision (von Platen et al., 2022). Inputs are resized to 640×320. Optimization uses Adam (Kingma & Ba, 2014) ($1 \times 10^{-5}$), with staged training: single-frame first (20 epochs, batch 160), then video sequences (batch 16) over three stages (40k/20k/10k steps).

## 4.2 QUANTITATIVE COMPARISON OF DEPTH COMPLETION

**Zero-shot Depth Completion.** As shown in Table 1 and 7, our approach achieves superior AbsRel and $\delta_1$ in most cases and consistently lower temporal error across all ScanNet TAE metrics. Compared to single-frame and video depth estimators, it retains competitive accuracy while preserving minimal temporal error. Relative to single-frame completion models, it demonstrates stronger cross-dataset consistency and temporal stability, confirming the efficacy of unified modeling in multi-domain zero-shot scenarios.

**Zero-shot Depth Inpainting.** We evaluate three degradation patterns: square masking, range truncation, and downsampling. Throughout Table 2 and 8, our method maintains optimal AbsRel and $\delta_1$ performance and reduced temporal instability on ScanNet TAE. Compared with single-frame completion models, it significantly reduces error in large occlusion and structural break cases without compromising temporal performance. Against video depth approaches, it sustains both low error and TAE under complex motion and heavy occlusion, demonstrating joint preservation of completion fidelity and inter-frame coherence across diverse masks.

Table 3: **Zero-shot Depth Completion with Mixed Prior.** ScanNet-TAE reports TAE↓, all other results are reported in AbsRel↓. "C": average result for random square masks; "D": applying 8× downsampling to the GT depths; "E": 1000(D+E) / 2000(E+C) random points.

| Model | KITTI | | | NYUv2 | | | Bonn | | | Sintel | | | ScanNet | | | ScanNet-TAE | | |
|---|---|---|---|---|---|---|---|---|---|---|---|---|---|---|---|---|---|---|
| | D+C | D+E | E+C | D+C | D+E | E+C | D+C | D+E | E+C | D+C | D+E | E+C | D+C | D+E | E+C | D+C | D+E | E+C |
| DAV2 | 13.18 | 13.18 | 12.72 | 28.95 | 28.91 | 29.25 | 8.44 | 8.35 | 8.33 | 49.9 | 45.67 | 43.65 | 12.65 | 12.65 | 12.66 | 2.243 | 2.249 | 2.253 |
| Depth pro | 9.92 | 10.01 | 9.95 | 20.08 | 20.29 | 19.75 | 6.4 | 6.37 | 6.4 | 132.65 | 129.4 | 126.84 | 8.76 | 8.75 | 8.72 | 3.322 | 3.317 | 3.334 |
| DepthCrafter | 10.75 | 10.92 | 11.25 | 28.65 | 28.29 | 28.52 | 6.19 | 6.51 | 6.33 | 47.5 | 41.46 | 34.12 | 13.24 | 13.16 | 13.14 | 1.857 | 1.905 | 1.846 |
| ChronoDepth | 17.37 | 17.54 | 17.7 | 34.61 | 34.35 | 34.57 | 8.16 | 8.04 | 8 | 70.96 | 70.5 | 70.89 | 16.43 | 16.37 | 16.33 | 1.524 | 1.527 | 1.527 |
| Depth Any Video | 8.57 | 8.9 | 8.03 | 17.23 | 17.1 | 16.66 | 7.51 | 7.25 | 7.36 | 46.47 | 42.97 | 42.18 | 10.59 | 10.66 | 10.39 | 1.881 | 1.891 | 1.879 |
| Video DA | 8.67 | 8.64 | 7.96 | 18.99 | 18.82 | 19.51 | 5.35 | 5.38 | 5.15 | 55.5 | **36.15** | 34.59 | 8.19 | 8.18 | 8.22 | 1.194 | 1.196 | 1.2 |
| DepthLab | 29.95 | 30.99 | 26.74 | 9.12 | 9.37 | 9.47 | 3.54 | 4.75 | 4.65 | 242.23 | 236.51 | 140.02 | 3.3 | 3.62 | 3.19 | 2.696 | 2.485 | 2.93 |
| Marigold-DC | 5.89 | 6.78 | 6.54 | 9.9 | 9.96 | 8.84 | 2.83 | 2.87 | 1.3 | 71.74 | 72.25 | 57.09 | 3.23 | 3.21 | 2.55 | 1.51 | 1.543 | 1.48 |
| Omni-DC | 5.65 | **5.52** | 5.63 | 9.41 | 9.4 | 9.32 | **1.72** | 2.66 | **1.12** | 46.73 | 46.36 | 39.16 | 2.84 | **2.4** | 2.39 | 1.332 | 1.16 | 1.337 |
| PriorDA | **5.12** | 6.05 | 5.56 | **8.83** | 8.68 | 8.65 | 3.02 | 2.97 | 1.67 | 50.2 | 54.8 | 50.29 | **2.53** | 2.77 | **1.91** | 1.181 | 1.07 | 1.174 |
| UniVDC(ours) | 6.82 | 5.98 | 4.92 | 9.02 | 8.55 | 8.59 | 2.12 | 2.21 | 1.88 | **45.78** | 40.88 | 32.43 | 2.74 | 2.64 | 2.38 | **0.867** | 0.84 | 0.843 |

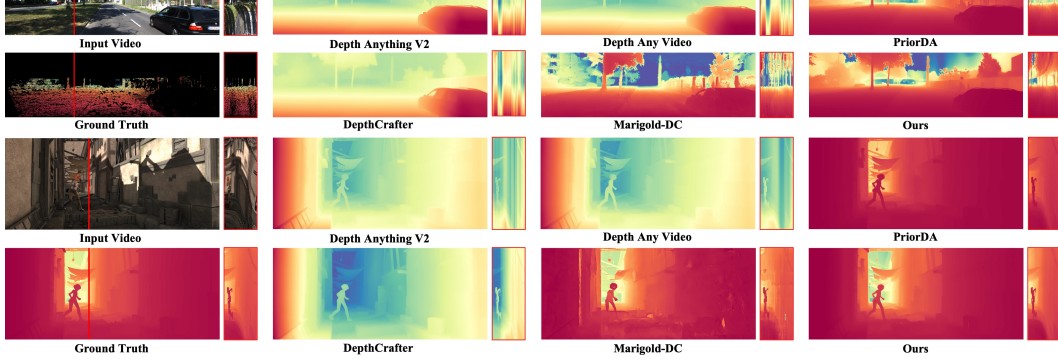

Figure 4: **Qualitative Comparison for Video Depth Completion.** For better visualizing the temporal quality, we show the temporal profiles of each result in red boxes, by slicing the depth values along the time axis at the red line positions.

**Zero-shot Depth Completion with Mixed Prior.** Table 3 and 9 demonstrate fused inputs of sparse, downsampled, and point-sampled data. Our method consistently achieves superior AbsRel and $\delta_1$ across datasets and prior combinations, with minimal ScanNet TAE error. Compared to other categories, it maintains enhanced scale stability and cross-scene consistency under multi-prior fusion, confirming generalization trends while reflecting effective prior utilization. This unified framework simultaneously adapts to complex sparse inputs and preserves temporal stability.

## 4.3 QUALITATIVE COMPARISON OF DEPTH COMPLETION

Figure 4 compares temporal consistency of state-of-the-art methods on KITTI and Sintel datasets, using 8-beam LiDAR and ORB/SIFT features respectively as sparse priors. Visualization analyses reveal that depth completion methods achieve superior scale accuracy and structural integrity over depth estimation approaches. Our technique further demonstrates enhanced structural completeness and smoothness compared to alternative completion methods. Temporal slices exhibit significant flickering and scale drift in estimation techniques. Although Marigold-DC and PriorDA closely approach ground truth, their single-frame limitations result in persistent flickering and discontinuous transitions.

Figure 5 further compares inter-frame consistency between PriorDA and our method on KITTI. Green rectangles highlight PriorDA's pronounced flickering and incomplete depth recovery between frames. This evidence confirms that even advanced single-frame completion methods remain constrained by per-frame priors, failing to leverage sequential temporal information for enhanced depth reconstruction.

Table 4 shows that our model attains leading accuracy while maintaining competitive inference speed. Expanding the sliding window from the baseline to longer sequences (supporting bidirec-

Table 4: **Performance and inference efficiency comparisons** on the ScanNet dataset (90 frames). Ours: 25-frame window with 15 overlapping frames; Ours-50: 50-frame window (25 overlapping); Ours-50-uni: unidirectional 50-frame window; Ours-100: 100-frame window. (On Nvidia H20).

| Model | DAV2 | Depth pro | Depth Crafter | Chrono Depth | Depth AV | Video DA | Depth Lab | Marigold DC | Omni DC | Prior DA | Ours | Ours 50 | Ours 50-Uni | Ours 100 |
|---|---|---|---|---|---|---|---|---|---|---|---|---|---|---|
| Paramters / M | 335 | 952 | 2157 | 1525 | 1423 | 382 | 2080 | 1290 | 85 | 433 | 2492 | 2492 | 2492 | 2492 |
| Runtime / s | **4.981** | 63.182 | 28.404 | 30.437 | 11.487 | 5.244 | 2951.763 | 9497.182 | 109.178 | 58.587 | 66.517 | 47.381 | 36.975 | 29.037 |
| AbsRel↓ | 13.028 | 9.037 | 13.6 | 16.61 | 10.856 | 8.549 | 5.243 | 4.117 | 3.886 | 3.888 | **3.664** | 3.788 | 4.32 | 9.224 |
| $\delta_1$↑ | 82.832 | 92.269 | 81.814 | 76.090 | 88.563 | 92.862 | 94.321 | 94.893 | 95.640 | 95.340 | **98.780** | 98.670 | 97.272 | 87.743 |
| TAE↓ | 2.208 | 3.268 | 1.820 | 1.503 | 1.860 | 1.186 | 3.042 | 1.866 | 1.841 | 1.452 | **0.921** | 0.937 | 1.053 | 1.579 |

Table 5: **Quantitative Aablation Studies.** We investigate the impact of different training and inference strategies on model performance. (w/o) indicates that the latents of the overlapping parts do not add noise.

| | Stage 1 | Stage 2 | Stage 3 | Stage 4 | Naive | Unidir.(w/o) | Unidir. | Bidir.(w/o) | Bidir. | KITTI AbsRel ↓ | KITTI $\delta_1$ ↑ | ScanNet AbsRel ↓ | ScanNet $\delta_1$ ↑ | TAE ↓ |
|---|---|---|---|---|---|---|---|---|---|---|---|---|---|---|
| (A) | ✓ | | | | ✓ | | | | | 15.72 | 80.18 | 12.72 | 81.38 | 2.253 |
| (B) | | ✓ | ✓ | ✓ | ✓ | | | | | 9.95 | 91.93 | 7.84 | 87.14 | 1.511 |
| (C) | ✓ | ✓ | | ✓ | ✓ | | | | | 8.98 | 92.78 | 6.93 | 90.03 | 1.589 |
| (D) | ✓ | ✓ | ✓ | | ✓ | | | | | 9.45 | 92.04 | 7.45 | 88.36 | 1.491 |
| (E) | ✓ | ✓ | ✓ | ✓ | ✓ | | | | | 8.03 | 94.59 | 5.97 | 93.14 | 1.437 |
| (F) | ✓ | ✓ | ✓ | ✓ | | ✓ | | | | 7.03 | 95.05 | 4.55 | 96.81 | 1.061 |
| (G) | ✓ | ✓ | ✓ | ✓ | | | ✓ | | | 6.88 | 95.14 | 4.32 | 97.27 | 1.053 |
| (H) | ✓ | ✓ | ✓ | ✓ | | | | ✓ | | 5.99 | 96.91 | 3.84 | 98.52 | 0.942 |
| (I) | ✓ | ✓ | ✓ | ✓ | | | | | ✓ | **5.84** | **97.89** | **3.66** | **98.78** | **0.921** |

tional and unidirectional modes) markedly increases throughput and accelerates inference, with only controlled accuracy degradation that remains within a competitive range. The baseline configuration leads in accuracy without lagging in speed among comparable methods. A medium-sized window further improves speed with near-optimal accuracy. Unidirectional inference reduces latency while preserving temporal stability. An ultra-long window achieves higher speed at the cost of a moderate accuracy drop. With tunable window length and overlap, the unified diffusion framework robustly balances efficiency and accuracy even at large parameter scales.

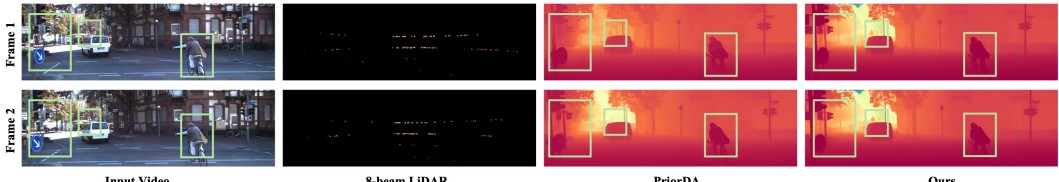

Figure 5: **Inter-Frame Consistency Comparison.** Additional results are provided in Appendix A.4. Additional experimental results are demonstrated in Part 1 of the supplementary video.

## 4.4 ABLATION STUDIES

**4-stage training strategy (A–E).** Table 5 demonstrates complementary effects across training stages. Training solely on single-frame data (A vs.B) yields the worst performance in both completion accuracy and temporal consistency. Omitting Stage 3 training (C vs.B vs.E) causes marginal degradation in completion performance but severe deterioration in temporal consistency, thus confirming the critical role of long-term temporal optimization for enhancing temporal awareness. When Stage 4 training is excluded (D vs.B vs.E), completion accuracy declines markedly, demonstrating that enhanced spatial perception directly improves completion efficacy.

**Bidirectional overlapping sliding-window inference (E–I).** As Table 5 shows, compared to naive inference (E), overlapping sliding windows significantly enhance both completion accuracy and temporal consistency. When injecting noise into denoised latents for re-optimization (F vs.G, H vs.I), both metrics improve further, demonstrating that noise refinement optimizes latent representations with known priors. Bidirectional sliding windows (BOSW) (H, I) outperform unidirectional variants (F, G) in completion accuracy and temporal consistency. As demonstrated in Figure 6 using 8-beam LiDAR depth priors, our bidirectional inference framework effectively mitigates scale drift for fast-moving objects while reducing depth flickering. Simultaneously (row 2), it suppresses error accumulation during prior propagation, validating BOSW's efficacy in enhancing temporal coherence.

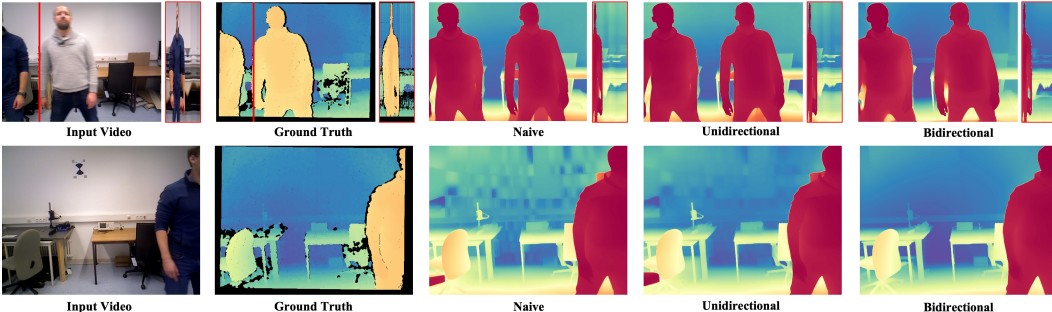

Figure 6: **Qualitative Comparison of Inference Strategies.** Bottom: last frame of the video clip.

Table 6: **Ablation Study on Input Conditions.** Effects of relative-depth backbone selection, prior noise injection, and RGB conditioning on UniVDC. We compare different MDE backbones for $d_{rel}$ (DAV2-B/DepthPro/DAV2-L), noise perturbations to three priors ($d_{rel}/d_c$/RGB), and two RGB conditioning pathways (CLIP semantics vs. direct images). Metrics are reported on KITTI and ScanNet as AbsRel↓/$\delta_1$↑/TAE↓. Perturbation methodologies are detailed in Appendix A.6.

| | $d_{rel}$ Base Model | | | | Perturbation Injection | | | RGB Image Injection | | KITTI | | ScanNet | | |
| | w/o $d_{rel}$ | DAV2-B | DepthPro | DAV2-L | $d_{rel}$ | $d_c$ | RGB | CLIP | Dire. | AbsRel ↓ | $\delta_1$ ↑ | AbsRel ↓ | $\delta_1$ ↑ | TAE ↓ |
|---|---|---|---|---|---|---|---|---|---|---|---|---|---|---|
| (J) | ✓ | | | | | | | ✓ | | 7.14 | 89.21 | 5.31 | 88.96 | 1.084 |
| (K) | | ✓ | | | | | | ✓ | | 6.15 | 93.87 | 3.85 | 93.84 | 0.969 |
| (L) | | | ✓ | | | | | ✓ | | 5.92 | **98.03** | **3.62** | 98.75 | **0.912** |
| (M) | | | | ✓ | | | | ✓ | | **5.84** | 97.89 | 3.66 | **98.78** | 0.921 |
| (N) | | | | ✓ | ✓ | | | ✓ | | 6.21 | 93.47 | 3.91 | 93.99 | 0.970 |
| (O) | | | | ✓ | | ✓ | | ✓ | | 6.49 | 88.10 | 4.47 | 87.90 | 1.023 |
| (P) | | | | ✓ | | | ✓ | ✓ | | 6.01 | 97.79 | 3.75 | 97.84 | 0.923 |
| (Q) | | | | ✓ | ✓ | ✓ | ✓ | ✓ | | 6.87 | 83.21 | 5.31 | 83.96 | 1.154 |
| (R) | | | | ✓ | | | | | ✓ | 6.97 | 88.18 | 5.07 | 88.91 | 1.023 |
| (S) | | | | ✓ | | | ✓ | | ✓ | 7.89 | 81.16 | 6.21 | 79.97 | 1.359 |
| (T) | | | | ✓ | | | | ✓ | ✓ | / | / | / | / | / |

**Ablation Study on Input Conditions.** As Table 6 shows, the results substantiate that UniVDC is a unified spatiotemporal diffusion framework leveraging multiple priors rather than a post-hoc refinement of a single estimator. First, replacing the $d_{rel}$ source across MDE backbones (DepthPro, DAV2-L) consistently improves AbsRel and TAE relative to w/o $d_{rel}$, confirming that ordinal cues function as one component within a multi-prior conditioning scheme rather than a fixed dependency (J, K, L, M). Notably, even without $d_{rel}$, UniVDC remains competitive against alternative methods, indicating that metric anchors ($d_c$), semantic conditioning, and temporal modeling (BOSW + staged training) provide substantial performance and stability on their own (J, vs. Table 1/2/3 baselines). Second, the three perturbations ($d_c$, $d_{rel}$, RGB) affect the model to different degrees; even under the most severe case with all three applied simultaneously, UniVDC remains competitive, highlighting the resilience of its multi-prior conditioning and temporal modeling (N, O, P, Q). Third, when using direct RGB as the conditioning input, the model becomes highly sensitive to RGB perturbations, whereas CLIP-based semantic conditioning is markedly more robust in maintaining cross-domain and long-range consistency (R, S vs.M). Together with staged training and bidirectional overlap inference, these ablations demonstrate that gains arise from unified diffusion conditioning over multi-source priors and temporal modeling, not from reliance on any single external estimator.

## 5 CONCLUSION

UniVDC establishes the first zero-shot unified framework for open-world video depth completion. It transforms incomplete and degraded depth inputs augmented by monocular cues into metrically consistent, temporally coherent sequences. By integrating geometric priors through a conditioned spatiotemporal diffusion process with four-stage training and bidirectional sliding window inference, the framework unifies multi-degradation adaptation, long-term consistency, and open-domain generalization. Experiments demonstrate comprehensive improvements in sparse depth accuracy, structural fidelity, scale stability, and temporal smoothness across variable-length videos. We anticipate UniVDC will advance downstream video-centric applications through enhanced geometric understanding.

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

## A APPENDIX

### A.1 ETHICS STATEMENT

We adhere to the ICLR Code of Ethics. All authors have read and are committed to complying with its guidelines throughout this research.

### A.2 REPRODUCIBILITY STATEMENT

To ensure reproducibility, we will release the implementation of our algorithm in an open-source repository upon acceptance of this paper.

### A.3 THE USE OF LARGE LANGUAGE MODELS (LLMS)

We use LLM to polish writing.

### A.4 INTER-FRAME CONSISTENCY COMPARISON

Comprehensive inter-frame consistency comparisons between our method and PriorDA are provided in Figure 7. Comparative results of all baselines under varying sparse depth patterns are available in the supplementary video (UniVDC_Video.mp4).

### A.5 QUANTITATIVE COMPARISON OF DEPTH COMPLETION

Tables 7, 8 and 9 present the $\delta_1$ metrics of depth completion performance for each method under varied sparse depth patterns.

### A.6 PERTURBATION INJECTION.

- Depth perturbation ($d_c$, $d_{rel}$): Gaussian noise addition: ±5% per depth value.
- Image perturbation (RGB):
    - Gaussian noise: $\mu = 0, \sigma = 1$
    - Rotation: $[0°, 360°]$
    - Translation: ±50 pixels
    - Horizontal flipping
    - Random rectangular occlusion (50px - image width)

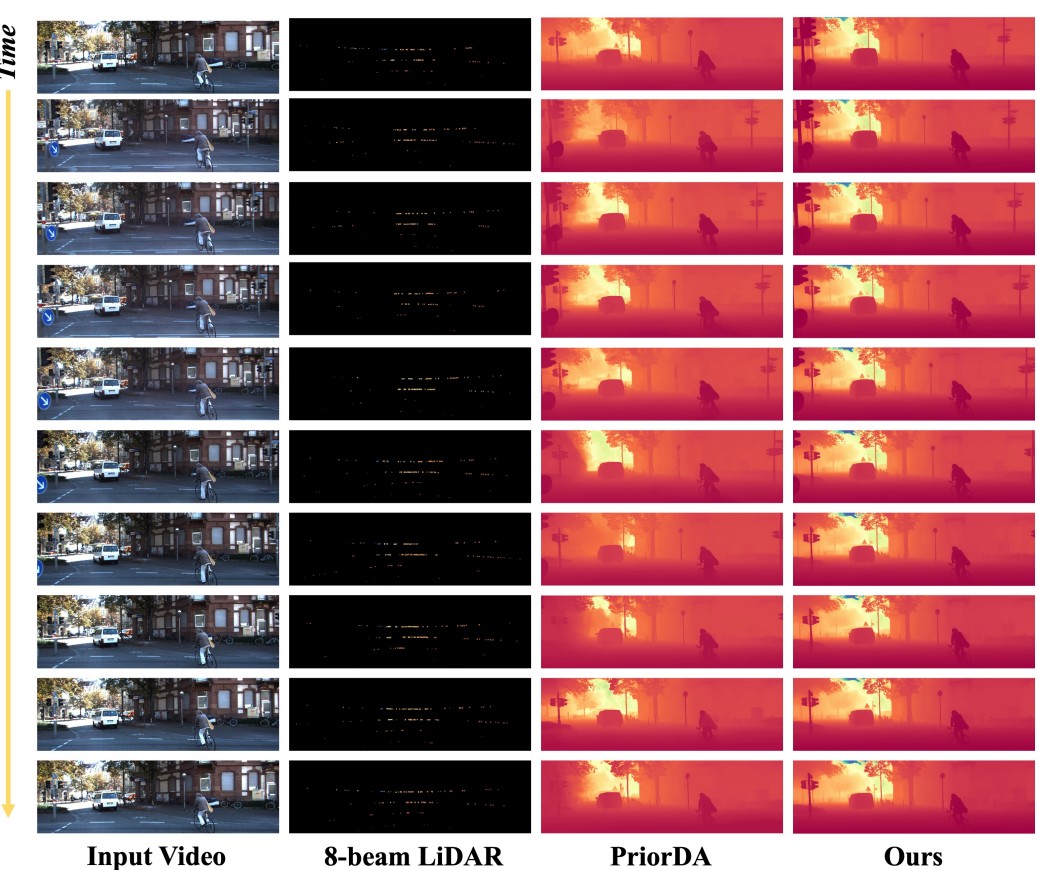

Figure 7: Comprehensive Inter-Frame Consistency Comparison.

Table 7: **Zero-shot Depth Completion.** ScanNet-TAE reports TAE↓, all other results are reported in $\delta_1$↑. "S": points sampled with SIFT and ORB; "E": 100 random points; "L": 8 LiDAR lines.

| Model | KITTI | | | NYUv2 | | | Bonn | | | Sintel | | | ScanNet | | | ScanNet-TAE | | |
|---|---|---|---|---|---|---|---|---|---|---|---|---|---|---|---|---|---|---|
| | S | E | L | S | E | L | S | E | L | S | E | L | S | E | L | S | E | L |
| DAV2 | 83.08 | 84.24 | 84.3 | 45.44 | 44.5 | 45.19 | 93.93 | 94.17 | 94.65 | 52.62 | 53.07 | 52.44 | 82.45 | 84.36 | 84.38 | 2.217 | 2.254 | 2.253 |
| Depth pro | 87.20 | 92.08 | 91.59 | 57.09 | 69.43 | 69.18 | 97.6 | 97.73 | 97.97 | 53.05 | 52.91 | 52.52 | 91.83 | 93.17 | 93.4 | 3.207 | 3.328 | 3.321 |
| DepthCrafter | 86.81 | 88.89 | 88.58 | 48.86 | 44.52 | 45.52 | 97.16 | 97.05 | 97.51 | 67.71 | 65.98 | 65.14 | 82.74 | 83.33 | 83.57 | 1.776 | 1.91 | 1.829 |
| ChronoDepth | 63.81 | 73.04 | 71.43 | 44.1 | 45.32 | 45.4 | 94.32 | 94.42 | 94.84 | 55.41 | 53.78 | 55.27 | 77.13 | 77.71 | 78.16 | 1.532 | 1.525 | 1.528 |
| Depth Any Video | 91.07 | 94.66 | 94.76 | 74.4 | 74.19 | 73.96 | 95.02 | 94.91 | 97.2 | 62.86 | 63.19 | 62.99 | 87.69 | 89.59 | 89.72 | 1.763 | 1.898 | 1.885 |
| Video DA | 94.89 | 95.06 | 94.66 | 68.96 | 66.69 | 70.27 | 97.4 | 97.56 | 94.48 | 63.86 | 62.31 | 63.09 | 92.32 | 93.88 | 93.9 | 1.171 | 1.199 | 1.198 |
| DepthLab | 48.21 | 34.98 | 33.42 | 91.29 | 88.47 | 89.8 | 94.7 | 93.81 | 95.45 | 61.27 | 49.67 | 58.95 | 92.56 | 95.16 | 96.46 | 4.718 | 5.33 | 3.397 |
| Marigold-DC | 95.39 | 95.56 | 94.41 | 89.62 | 90.19 | 90.2 | 99.13 | 98.54 | 98.89 | 76.67 | 82.2 | 83.29 | 96.31 | 97.15 | 97.18 | 2.899 | 2.439 | 1.774 |
| Omni-DC | 96.92 | 96.22 | 95.06 | 91.55 | 91.43 | 91.6 | 99.02 | 98.3 | 98.85 | 93.38 | 86.57 | 93.45 | 96.56 | 98.00 | 98.4 | 2.575 | 2.621 | 1.545 |
| PriorDA | 98.01 | 97.52 | 97.05 | 89.69 | 89.88 | 89.76 | 98.85 | 98.77 | 98.91 | 88.21 | 87.65 | 89.21 | 97.08 | 97.73 | 97.64 | 1.991 | 1.783 | 1.446 |
| UniVDC(ours) | 98.99 | 96.99 | 97.61 | 92.33 | 91.93 | 90.92 | 98.51 | 98.83 | 98.43 | 94.15 | 88.31 | 94.35 | 98.77 | 98.67 | 98.73 | 1.032 | 0.875 | 0.928 |

Table 8: **Zero-shot Depth Inpainting.** ScanNet-TAE reports TAE↓, all other results are reported in $\delta_1$↑. "C": average result for random square masks; "R": masks for depth beyond 2m (indoors) and 15m (outdoors); "D": applying 8× downsampling to the GT depths.

| Model | KITTI | | | NYUv2 | | | Bonn | | | Sintel | | | ScanNet | | | ScanNet-TAE | | |
|---|---|---|---|---|---|---|---|---|---|---|---|---|---|---|---|---|---|---|
| | C | R | D | C | R | D | C | R | D | C | R | D | C | R | D | C | R | D |
| DAV2 | 84.24 | 76.97 | 82.72 | 44.61 | 26.35 | 44.76 | 94.42 | 57.36 | 94.43 | 52.75 | 51.04 | 53.56 | 84.34 | 72.41 | 84.4 | 2.26 | 1.898 | 2.244 |
| Depth pro | 92.42 | 79.73 | 91.71 | 69.87 | 33.98 | 69.37 | 97.78 | 69.01 | 97.79 | 52.74 | 49.45 | 53.02 | 93.17 | 86.44 | 93.11 | 3.332 | 2.932 | 3.318 |
| DepthCrafter | 88.24 | 85.75 | 87.18 | 43.69 | 23.49 | 43.86 | 97.31 | 64.45 | 97.51 | 65.05 | 57.76 | 64.48 | 83.48 | 70.34 | 82.9 | 1.898 | 1.528 | 1.828 |
| ChronoDepth | 73.46 | 69.74 | 71.46 | 45.33 | 24.34 | 45.37 | 94.59 | 68.18 | 94.69 | 53.76 | 52.55 | 54.05 | 77.7 | 63.92 | 77.58 | 1.527 | 1.315 | 1.525 |
| Depth Any Video | 94.38 | 82.34 | 93.37 | 75.19 | 39.41 | 75.52 | 95.39 | 62.74 | 95.12 | 64.51 | 61.08 | 64.55 | 89.74 | 81.25 | 89.99 | 1.887 | 1.782 | 1.876 |
| Video DA | 95.07 | 80.66 | 93.08 | 66.04 | 33.62 | 67.91 | 97.22 | 70.21 | 97.24 | 62.35 | 57.8 | 62.96 | 93.84 | 86.32 | 93.87 | 1.199 | 1.127 | 1.193 |
| DepthLab | 92.74 | 75.01 | 52.37 | 91.92 | 49.23 | 92.49 | 98.92 | 47.66 | 98.54 | 86.58 | 78.9 | 66.02 | 98.61 | 74.89 | 97.97 | 2.028 | 2.055 | 1.742 |
| Marigold-DC | 99.27 | 80.08 | 97.83 | 92.16 | 48.54 | 91.26 | 99.61 | 49.68 | 98.91 | 85.3 | 80.23 | 81.59 | 97.67 | 76.74 | 97.15 | 1.398 | 2.451 | 1.296 |
| Omni-DC | 98.93 | 79.1 | 98.69 | 91.73 | 49.26 | 92.08 | 99.27 | 51.07 | 99.01 | 92.55 | 80.33 | 95.23 | 98.39 | 76.1 | 98.53 | 2.519 | 2.545 | 0.931 |
| PriorDA | 98.75 | 74.81 | 98.41 | 90.9 | 48.21 | 91.07 | 99.1 | 48.00 | 98.99 | 93.34 | 80.6 | 91.37 | 97.84 | 75.16 | 97.94 | 1.212 | 2.302 | 0.908 |
| UniVDC(ours) | 99.22 | 92.83 | 98.77 | 93.08 | 89.36 | 93.22 | 99.33 | 70.73 | 99.14 | 95.97 | 80.38 | 93.11 | 98.97 | 98.54 | 98.71 | 0.932 | 1.115 | 0.856 |

Table 9: **Zero-shot Depth Completion with Mixed Prior.** ScanNet-TAE reports TAE↓, all other results are reported in $\delta_1$↑. "C": average result for random square masks; "D": applying 8× downsampling to the GT depths; "E": 1000(D+E) / 2000(E+C) random points.

| Model | KITTI | | | NYUv2 | | | Bonn | | | Sintel | | | ScanNet | | | ScanNet-TAE | | |
|---|---|---|---|---|---|---|---|---|---|---|---|---|---|---|---|---|---|---|
| | D+C | D+E | E+C | D+C | D+E | E+C | D+C | D+E | E+C | D+C | D+E | E+C | D+C | D+E | E+C | D+C | D+E | E+C |
| DAV2 | 82.65 | 82.64 | 84.18 | 44.75 | 44.76 | 44.31 | 94.41 | 94.28 | 94.15 | 53.55 | 53.55 | 52.77 | 84.4 | 84.39 | 84.36 | 2.243 | 2.249 | 2.253 |
| Depth pro | 91.88 | 91.65 | 91.93 | 69.59 | 69.08 | 70.42 | 97.78 | 97.79 | 97.72 | 53.02 | 53.07 | 52.9 | 93.07 | 93.08 | 93.15 | 3.322 | 3.317 | 3.334 |
| DepthCrafter | 87.64 | 87.83 | 88.03 | 45.33 | 46.38 | 46.37 | 97.72 | 96.94 | 97.34 | 65.8 | 65.93 | 65.91 | 83.12 | 83.11 | 83.74 | 1.857 | 1.905 | 1.846 |
| ChronoDepth | 71.87 | 71.27 | 97.36 | 45.27 | 43.38 | 45.27 | 94.67 | 94.58 | 94.4 | 53.93 | 54.13 | 53.75 | 77.45 | 77.57 | 77.59 | 1.524 | 1.527 | 1.527 |
| Depth Any Video | 94.14 | 93.98 | 94.59 | 73.38 | 74.12 | 75.94 | 94.66 | 94.91 | 94.82 | 63.47 | 63.15 | 64.33 | 89.53 | 89.6 | 89.96 | 1.881 | 1.891 | 1.879 |
| Video DA | 93.01 | 93.06 | 95.05 | 67.4 | 67.94 | 66.07 | 97.27 | 97.11 | 97.55 | 62.81 | 62.99 | 62.37 | 93.87 | 93.91 | 93.85 | 1.194 | 1.196 | 1.2 |
| DepthLab | 52.34 | 49.9 | 56.19 | 91.87 | 92.41 | 92.05 | 97.54 | 97.58 | 97.45 | 64.91 | 57.15 | 68.02 | 97.49 | 97.88 | 97.87 | 2.696 | 2.485 | 2.93 |
| Marigold-DC | 97.58 | 97.76 | 98.52 | 91.14 | 91.11 | 92.08 | 98.86 | 98.8 | 99.27 | 81.3 | 80.58 | 84.3 | 97.13 | 97.14 | 97.57 | 1.51 | 1.543 | 1.48 |
| Omni-DC | 98.4 | 98.64 | 99.11 | 91.95 | 92.15 | 92.2 | 98.9 | 98.99 | 99.31 | 94.89 | 94.11 | 96.48 | 98.44 | 98.59 | 97.75 | 1.332 | 1.16 | 1.337 |
| PriorDA | 98.27 | 98.39 | 98.61 | 90.69 | 90.64 | 90.73 | 98.94 | 98.94 | 98.97 | 90.95 | 90.61 | 91.97 | 97.85 | 97.91 | 98.91 | 1.181 | 1.07 | 1.174 |
| UniVDC(ours) | 98.95 | 98.36 | 99.29 | 92.09 | 92.34 | 92.22 | 99.11 | 98.96 | 99.33 | 95.05 | 95.19 | 93.93 | 98.81 | 98.97 | 98.88 | 0.867 | 0.84 | 0.843 |

